# Dendritic cell-elicited B-cell activation fosters immune privilege via IL-10 signals in hepatocellular carcinoma

Fang-Zhu Ouyang[1,2], Rui-Qi Wu[1], Yuan Wei[1], Rui-Xian Liu[1], Dong Yang[1], Xiao Xiao[1], Limin Zheng[1], Bo Li[3], Xiang-Ming Lao[2,*] & Dong-Ming Kuang[2,*]

B cells are prominent components of human solid tumours, but activation status and functions of these cells in human cancers remain elusive. Here we establish that over 50% B cells in hepatocellular carcinoma (HCC) exhibit an FcγRII$^{low/-}$ activated phenotype, and high infiltration of these cells positively correlates with cancer progression. Environmental semimature dendritic cells, but not macrophages, can operate in a CD95L-dependent pathway to generate FcγRII$^{low/-}$ activated B cells. Early activation of monocytes in cancer environments is critical for the generation of semimature dendritic cells and subsequent FcγRII$^{low/-}$ activated B cells. More importantly, the activated FcγRII$^{low/-}$ B cells from HCC tumours, but not the resting FcγRII$^{high}$ B cells, without external stimulation suppress autologous tumour-specific cytotoxic T-cell immunity via IL-10 signals. Collectively, generation of FcγRII$^{low/-}$ activated B cells may represent a mechanism by which the immune activation is linked to immune tolerance in the tumour milieu.

[1] Key Laboratory of Gene Engineering of the Ministry of Education, State Key Laboratory of Biocontrol, School of Life Sciences, Sun Yat-sen University, Guangzhou 510275, China. [2] State Key Laboratory of Oncology in Southern China, Collaborative Innovation Center for Cancer Medicine, Sun Yat-sen University Cancer Center, Guangzhou 510060, China. [3] Department of Biochemistry, Zhongshan School of Medicine, Sun Yat-sen University, Guangzhou 510080, China. * These authors jointly supervised this work. Correspondence and requests for materials should be addressed to X.-M.L. (email: laoxm@sysucc.org.cn) or to D.-M.K. (email: kdming@mail.sysu.edu.cn).

Tumour-promoting inflammation/immune activation and avoiding immune destruction have both emerged as hallmarks of human cancer[1–3]. Hepatocellular carcinoma (HCC) is usually present in inflamed fibrotic and/or cirrhotic liver with extensive leukocyte infiltration[4,5]. Thus, the immune status at a tumour site can largely influence the biological behaviour of HCC. High infiltration of immunosuppressive macrophages and regulatory T cells are both shown to correlate with reduced survival and increased invasiveness in HCC[6,7]. More strikingly, increased levels of activated monocytes and pro-inflammatory T helper 17 cells in HCC also predict poor prognosis[8,9]. Thus immune networks of human cancer environments are more complicated and heterogeneous than we have acknowledged and, in turn, suggest existence of unrecognized interaction/crosstalk between immune activation and immune suppression within cancer environments[10].

B cells consistently represent abundant cellular components in tumours, but the activation status and biological functions of B cells in human tumours are poorly understood[11]. In normal lymphoid organs, B cells express considerable suppressive receptor Fcγ receptor II (FcγRII; also termed CD32), but not FcγRI (CD64) or FcγRIII (CD16), to sustain immunoglobulin G-elicited inactivation of cells. Under the influence of inflammation, B cells actively downregulated FcγRII and promptly become activated in response to the environmental mediators[12]. Moreover, B-cell activation is often regulated by inflammatory cytokines, of which activated T-cell-derived IL-4 and IL-21 are the most effective[13,14]. In addition to being regulated by activated T cells, B-cell activation is also promoted by environmental antigen-presenting cells (APCs), particularly dendritic cells (DCs) and macrophages[15,16]. We have previously demonstrated that cancer environments induce formation of semimature DCs and dysfunctional macrophages[17,18]. However, at present, little is known about the regulation of DCs or macrophages on B-cell activation and functions in human tumours in situ. Thus, evaluating the FcγRII expression and the functional data on B cells in the presence of DCs or macrophages in cancer environments are essential for understanding their activation status and potential functions in tumour immunopathogenesis.

Studies in human peripheral blood have identified a subset of regulatory B cells that exhibit a CD24^highCD38^high phenotype, and it has been suggested that these cells play a crucial part in regulating T-cell responses by releasing IL-10 (refs 19,20). However, in this study, our results show that CD24^highCD38^high regulatory B cells are mainly detected in patients' blood, but not in paired HCC tumour. Instead, over 50% of the HCC-infiltrating B cells are FcγRII^low/− and display a CD69^+BTLA^− activated phenotype. More importantly, the activated FcγRII^low/− B cells, but not the resting FcγRII^high B cells, without additional stimulation are the major source of functional IL-10 production. We demonstrate that the early activation of monocytes induced by HCC environments is required for the sequential DC semimaturation, CD95L-elicited FcγRII^low/− activated B-cell generation, and IL-10-induced cytotoxic T-cell dysfunction. Therefore, induction of activated FcγRII^low/− IL-10-producing B cells by semimature DCs, which are differentiated from tumour-activated monocytes, may represent a mechanism by which the immune activation is linked to immune tolerance in the tumour milieu.

## Results

### Identification of activated FcγRII^low/− IL-10^+ B cells in HCC.
In paired blood and tumour tissues from HCC patients (n = 25), as well as blood from healthy donors (n = 10), the frequencies of

B cells in CD45^+ mononuclear cells ranged from 6 to 9% (Supplementary Fig. 1a,b). However, B cells in tumours in situ only selectively accumulated in the tumour-surrounding (peritumoral) stroma (Fig. 1a). B cells isolated from both normal (n = 5) and HCC blood samples were mainly inactivated and expressed significant suppressive receptor FcγRII (also termed CD32) (Fig. 1b). Interestingly, over 50% of the B cells from HCC tumours became activated and showed radical decrease of FcγRII. By contrast, the B cells from normal liver samples (tissue distal to a liver hemangioma, n = 2) showed higher FcγRII expression (Fig. 1b). FACS results confirmed the activated state of FcγRII^low/− B cells as following: most FcγRII^low/− B cells from HCC tumours displayed a CD69^+BTLA^− activated phenotype with reduced B-cell follicle homing molecules CD62L, CXCR5 and CCR6. B cells derived from paired HCC blood were mainly CD45RA^+IgD^+IgM^+IgG^−BTLA^+ (Fig. 1c). Unexpectedly, the proportions of activated FcγRII^low/− B cells in HCC tumour even positively correlated with patient's TNM stages (Fig. 1d). These findings prompted us to further investigate the functional features of HCC-infiltrating activated FcγRII^low/− B cells.

We purified the FcγRII^high and FcγRII^low/− B cells from HCC tumours. The purities of B cells we used were >98%, as assessed by determining the expression of myeloid cell marker CD33 and T-cell marker CD3 (Supplementary Fig. 1c). The FcγRII^low/− B cells, undergoing IL-21 plus CD40L stimulation, did not differentiate into immunoglobulin-secreting plasma cells (Fig. 1e), although they were activated. More abnormally, using an enzyme-linked immunospot (ELISpot) detection system, we observed that the FcγRII^low/− B cells, but not the FcγRII^high B cells, without additional stimulation, were the major source of IL-10 production in tumour B cells (Fig. 1f), which is in contrast to observations in mouse model that the FcγRII^high B cells were the major source of IL-10 production[16]. Consistently, B cells derived from mouse hepatoma models did not exhibit an FcγRII^low/− phenotype (Supplementary Fig. 1d). Notably, the CD24^highCD38^high B cells that were considered as conventional peripheral IL-10-producing B cells[19,20] were hardly detected in HCC tumours; and more importantly, without external stimulus, the CD24^highCD38^high B cells were unable to produce IL-10 (Supplementary Fig. 1e,f). These data together suggest that peritumoral environments of HCC tumours may activate B cells to adopt an FcγRII^low/− phenotype, which in turn endows the cells with functional production of protumorigenic IL-10.

### Tumour DC induces B-cell activation and IL-10 production.
Inasmuch as activated FcγRII^low/− B cells selectively distributed in HCC tumours (Fig. 1b), we next investigated the effects of HCC environments on activated FcγRII^low/− B-cell generation. APCs are critical for initiating and maintaining T- and B-cell immunity[21]. In HCC peritumoral stroma, the main site of B cells (Fig. 1a), there were pronounced accumulations of S100^+ DCs and CD68^+ macrophages (Fig. 2a,b and Supplementary Fig. 2a), and that increased densities of these cells in the peritumoral stroma both predicted reduced survival (Fig. 2c, Supplementary Table 1; ref. 8). Dissimilarly, S100^+ DCs in the nontumoral or intratumoral area of HCC tumours were unrelated to the prognosis (Fig. 2c). Multivariate analysis revealed that the number of S100^+ cells in peritumoral stroma of HCC was an independent prognostic factor of survival (Supplementary Table 2).

In subsequent experiments, we purified CD45^+CD15^− CD11c^highCD11b^high DCs (TDCs), as well as CD45^+CD15^− CD11c^lowCD11b^high macrophages (TAMs), from HCC tumours (Supplementary Fig. 2b), and then cultured those cells with autologous blood B cells ex vivo. TDCs displayed a semimature

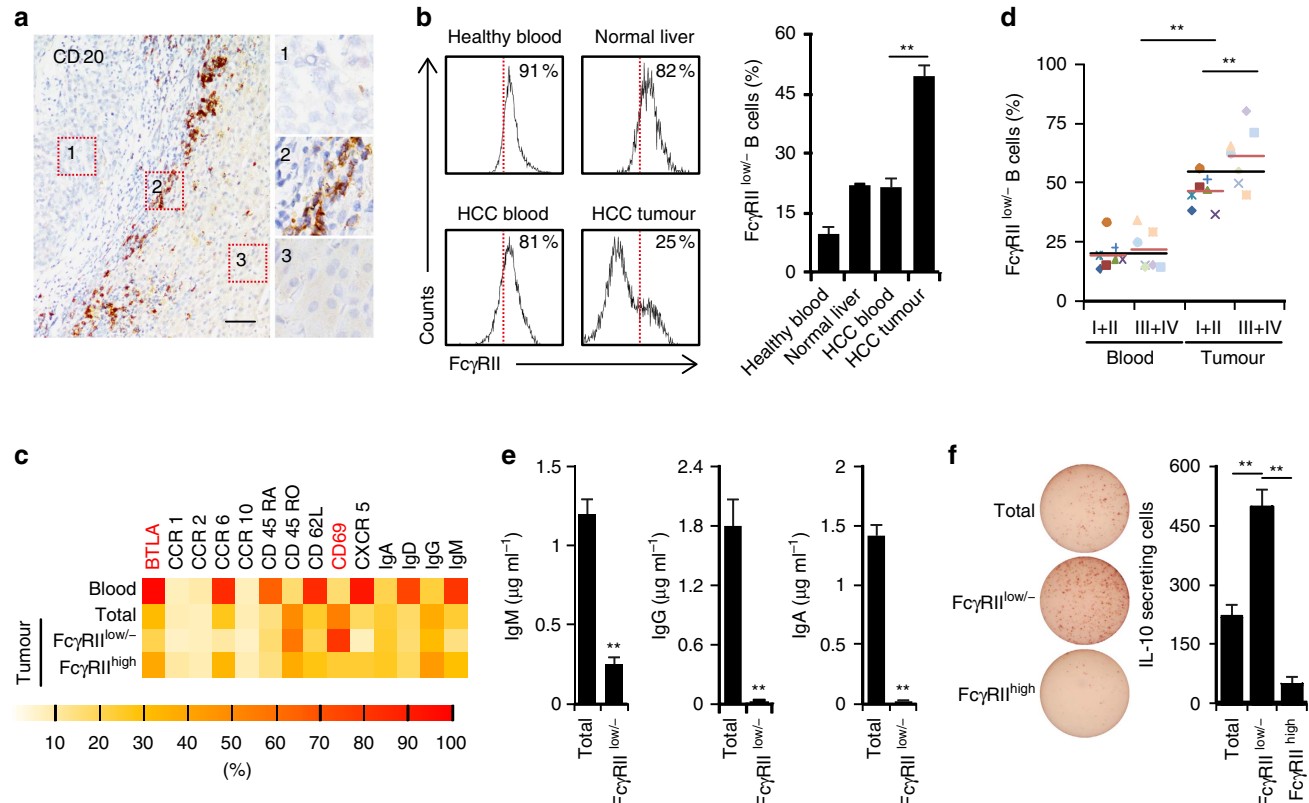

**Figure 1 | Accumulation of FcγRII^low/− B cells correlated with disease progression and functional IL-10 production in HCC tumours.** (a) Representative distribution of CD20+ B cells in intratumoral (1), peritumoral stromal (2) and nontumoral (3) areas of HCC samples ($n = 10$). Scale bar, 100 μm. (b) FACS analysis of FcγRII on CD19+ B cells from healthy blood ($n = 5$), normal liver ($n = 2$), and paired blood and HCC tumour ($n = 15$). (c) Phenotypic characteristics of B cells, FcγRII^low/− B cells or FcγRII^high B cells from paired blood and HCC tumour. Data for each of the indicated markers represent at least four patients. (d) Associations of tumour FcγRII^low/− B cells with patients' TNM stage are shown ($n = 7$ for stages I–II and $n = 7$ for stages III–IV). (e) B cells (total) and sorted FcγRII^low/− B cells from HCC tumours were stimulated with 1 μg ml−1 CD40L plus 50 ng ml−1 IL-21 for 6 days. IgM, IgG and IgA production were determined by ELISA. Results represent three independent experiments ($n = 5$). (f) IL-10 production by total, FcγRII^low/−, and FcγRII^high B cells from HCC tumours was determined by ELISpot ($n = 6$). *$P < 0.05$, **$P < 0.01$ (Student's $t$ test). Error bars, s.e.m.

phenotype, as shown by determining the expression of CD83, CD86 and HLA-DR (Fig. 2d), and they effectively downregulated B cell FcγRII and induced B-cell activation (Fig. 2e and Supplementary Fig. 2c,d). In the culture system of TDCs and B cells, we did not detect a marked apoptosis of B cells, as assessed by determining the number of dead trypan blue positive cells. In support of the above-mentioned findings that activated FcγRII^low/− B cells in tumours were more potent to produce IL-10 (Fig. 1f), B cells activated by TDCs also acquired capacity to spontaneously produce IL-10 (Fig. 2f). Conversely, TAMs isolated from HCC tumours did not elicit such B-cell activation and IL-10 production (Fig. 2e,f). Consistently, the number of S100+ DCs, but not that of CD68+ macrophages, positively correlated with the proportion of FcγRII^low/− B cells in HCC tumours (Fig. 2g). Notably, we found that immature DCs or mature DCs generated from healthy blood monocytes only weakly affected B-cell activation and subsequent IL-10 production (Supplementary Fig. 2e,f). Moreover, the FcγRII^low/− B cells induced by TDCs did not possess capability to differentiate into plasma cells (Supplementary Fig. 2g). Together, DCs generated in hepatoma environments is mainly crucial for B-cell activation-elicited IL-10 production.

**Monocyte activation regulates TDC formation and function.** The results described above suggested that HCC environments altered the normal development of monocyte-derived DCs, which

in turn triggered the activation-elicited generation of FcγRII^low/− IL-10-producing B cells. To address this possibility, we generated DCs from HCC tumour-isolated CD14+ monocytes (TMO-DCs) and paired blood CD14+ monocytes (BMO-DCs) (Supplementary Fig. 3a). Only the TMO-DCs, but not the BMO-DCs, exhibited a semimature phenotype (Fig. 3a) as that displayed by DCs directly isolated from HCC tumours (Fig. 2d). Accordingly, only the TMO-DCs successfully induced B-cell activation and IL-10 production (Fig. 3b,c). These findings suggest that the initial activation status of monocytes in tumours determines the final phenotypic and functional characteristics of DCs.

We have recently observed that hyaluronan fragments (HA) constitute a common factor produced by human tumours, including hepatoma, to induce transient early activation of monocytes[7]. Therefore, we next determined whether such a mechanism was also responsible for the formation of semimature DCs in tumour environments. As expected, pre-exposure of monocytes to culture supernatant from primary HCC cells (HCC-SN) for 1 h could result in formation of semimature DCs (Fig. 3d), which were able to trigger B-cell activation and IL-10 production (Fig. 3e,f). It is noteworthy that MAPKs and IκB pathways have been considered as important regulators for monocyte innate activation[22,23]. Analysing the dynamic signalling pathways revealed rapid activation of MAPKs Jnk, Erk and p38, and the NF-κB inhibitor IκBα in monocytes exposed to the primary HCC-SN (Fig. 3g). More interestingly, inhibiting the activation of IκBα, but not Jnk, Erk or p38, in HCC-SN-exposed

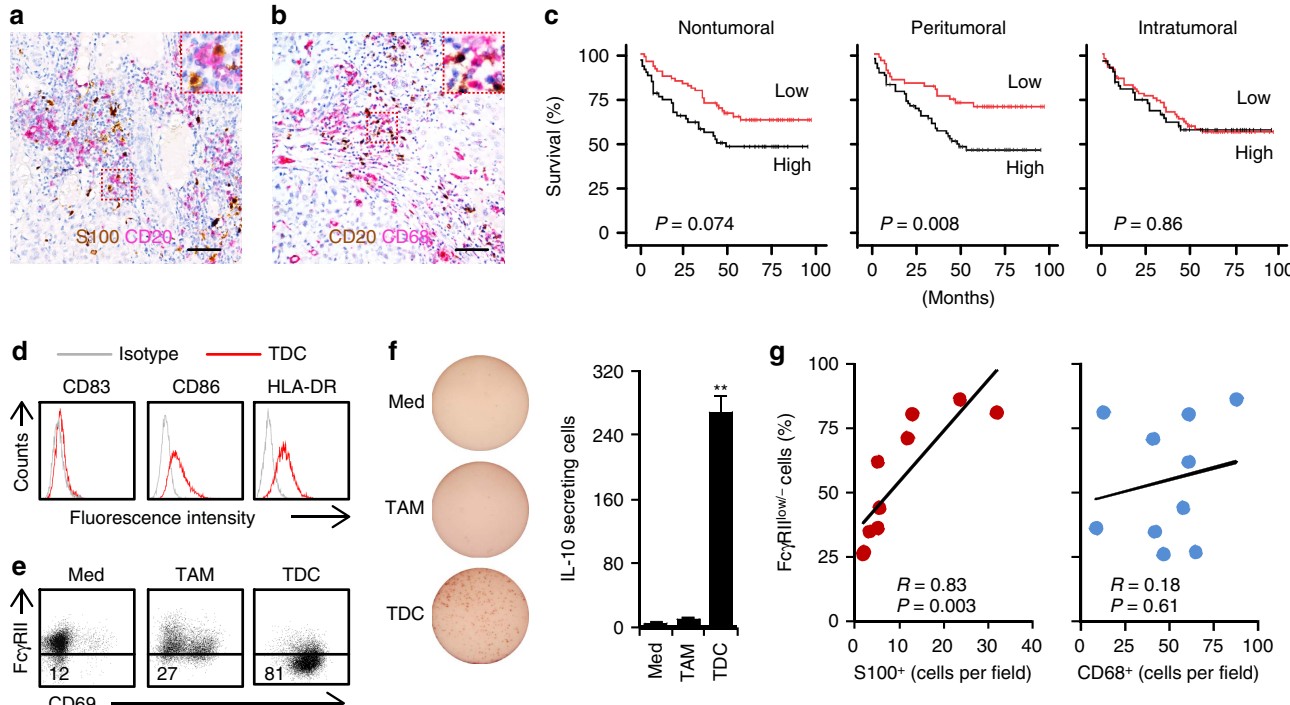

**Figure 2 | HCC tumour-derived DCs were required for IL-10-producing FcγRII$^{low/-}$ B-cell generation. (a,b)** Representative distribution of S100$^+$ DCs (brown) plus CD20$^+$ B cells (red) (**a**) or CD20$^+$ B cells (brown) plus CD68$^+$ macrophages (red) (**b**) in paraffin-embedded HCC samples ($n=10$). Scale bar, 100 μm. (**c**) Cumulative survival curves of patients. The patients ($n=135$) were divided into two groups according to the median value of S100$^+$ DC densities in nontumoral (median value $=0.5$), peritumoral stromal (median value $=20.5$) and intratumoral (median value $=0.2$) regions of HCC: red lines, low density; black lines, high density. (**d**) Expression of CD83, CD86 and HLA-DR in HCC tumour-derived CD11c$^{high}$CD11b$^{high}$ DCs (TDCs) were determined by FACS ($n=4$). (**e,f**) HCC tumour-derived TDCs, but not CD11c$^{low}$CD11b$^{high}$ macrophages (TAMs), effectively induced IL-10-producing FcγRII$^{low/-}$ B cells. Results represent three independent experiments ($n=5$). (**g**) Associations of tumour FcγRII$^{low/-}$ B cells with peritumoral stromal S100$^+$ DC and CD68$^+$ macrophages densities are shown ($n=10$). *$P<0.05$, **$P<0.01$ (Student's $t$-test). Error bars, s.e.m.

monocytes could significantly attenuate the sequential DC semimaturation, and B-cell activation and IL-10 production (Fig. 3h,i and Supplementary Fig. 3b). Thus, IκBα-mediated early activation of monocytes in hepatoma environments is vital for semimature DC-elicited B-cell activation and IL-10 production. In line with our hypothesis, pre-exposure of monocytes to HCC-SN for 1 h in the presence of a hyaluronan-specific blocking peptide (Pep-1) could partially inhibit the semimaturation of DCs (Fig. 3j), as well as subsequent generation of IL-10-producing FcγRII$^{low/-}$ B cells Fig. 3k,l).

**Role of CD95–CD95L axis in TDC-B-cell interaction.** It is known that DCs regulate B-cell responses via membrane bound molecules and secretion of soluble mediators[21]. To probe the underlying molecular mechanisms that allow TDCs to induce FcγRII$^{low/-}$ IL-10-producing B cells, TDCs were pretreated with mitomycin C ($10\,\mu g\,ml^{-1}$) to prevent the cytokine release (Supplementary Fig. 4a) and subsequently co-cultured with autologous peripheral B cells. Such treatment completely blocked the ability of TDCs to induce B-cell activation and IL-10 production (Fig. 4a), suggesting that soluble mediators released by TDCs may trigger the above-mentioned B-cell activation and IL-10 production. In support, although to a lesser extent, the conditioned medium from TDCs did promote B-cell activation and IL-10 production (Fig. 4b).

We afterward examined immune regulatory molecules expressed by HCC-SN-induced DCs. Compared with normal immature DCs, HCC-SN-induced semimature DCs expressed higher levels of IFN-β1 and IFN-β2, but not CD40L, COX-1,

COX-2, IDO-1 or IDO-2 (Supplementary Fig. 4b). We also detected significant amounts of IFN-β1 and IFN-β2 in the conditioned medium from TDCs (Supplementary Fig. 4c). Inasmuch as a recent study using mouse model has shown that CD40L acts synergistically with IFN-β to induce IL-10-producing B cells[16], we conducted experiments using a neutralizing Ab against CD40L, IFN-β1 and/or IFN-β2 to ascertain whether these molecules gave rise to FcγRII$^{low/-}$ IL-10-producing B cells in humans. However, usage of an Ab at a concentration that effectively neutralized IFN-β1 or IFN-β2 in the conditioned medium from TDCs did not reduce the generation of FcγRII$^{low/-}$ IL-10-producing B cells (Supplementary Fig. 4d–f), suggesting that distinct induction mechanisms of IL-10 production in B cells are employed by humans and mice.

We have previously observed that the semimature DCs generated in tumour environments primarily activated autologous T cells and subsequently led to apoptosis of those cells[17]. It is well known that CD95–CD95L interaction plays a crucial role in DC-mediated activation-induced T-cell death[24,25]. Thus, we analysed whether CD95–CD95L interaction participated in TDC-mediated activated FcγRII$^{low/-}$ IL-10-producing B-cell generation. In fact, in our TDC-B-cell culture system, a marked increase of CD95 was detected in B cells (Fig. 4c). Consistently, increased CD95 expression was also detected in tumour-derived B cells compared with blood B cells (Fig. 4d). Moreover, measuring the CD95L expression over time in DCs revealed gradual upregulation of CD95L in HCC-SN-induced semimature DCs, reaching a maximum or a plateau on day 5 (Fig. 4e). Similar upregulation of CD95L were also detected in TMO-DCs and TDCs (Fig. 4f). These findings provide further evidence that

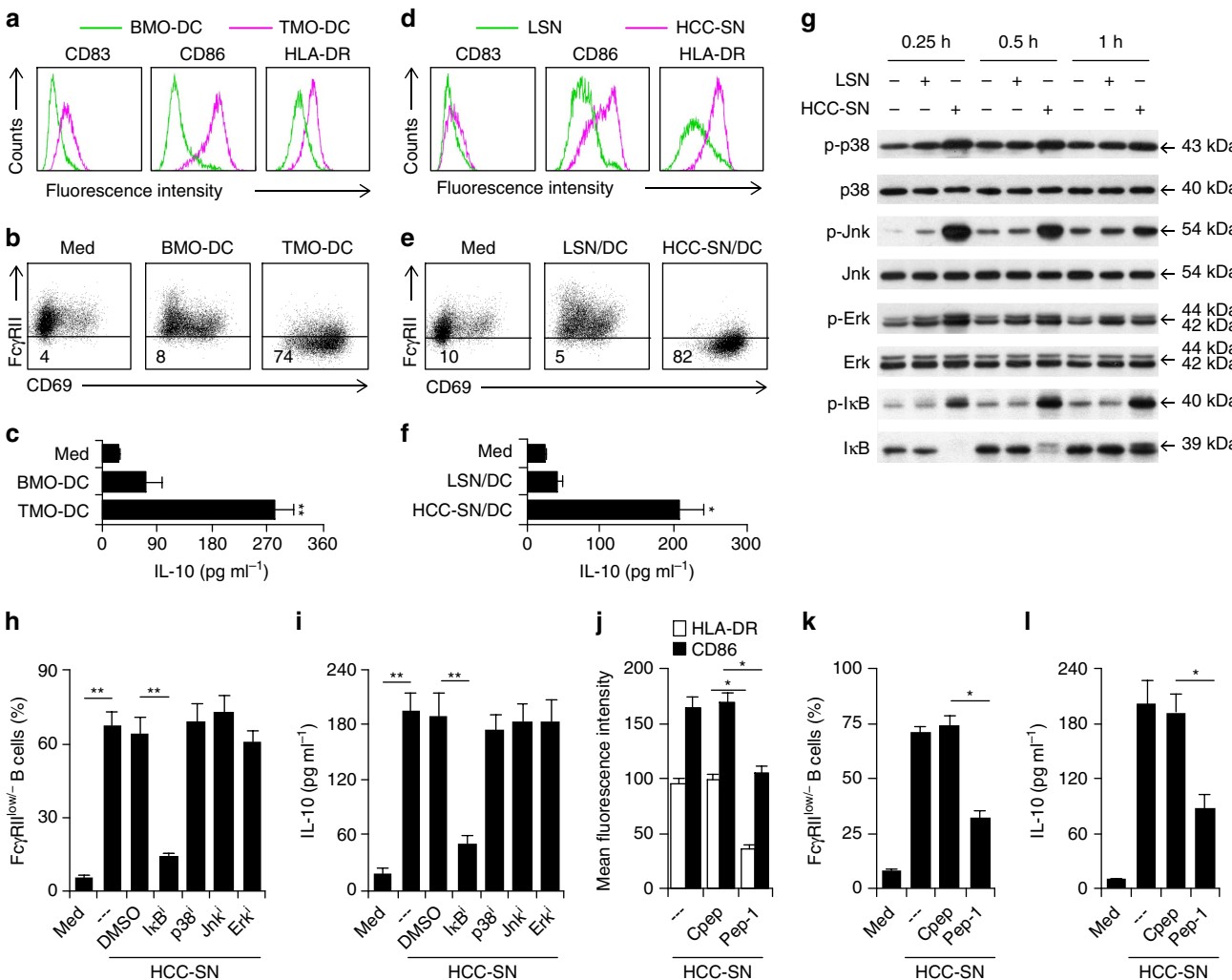

**Figure 3 | Roles of early activation of monocytes in tumour DC-elicited B-cell activation and subsequent IL-10 production.** (**a–f**) Peripheral B cells were cultured for 24 h in medium (Med) or with autologous blood or tumour monocyte-derived DCs (BMO-DC or TMO-DC) (**a–c**); peripheral B cells were cultured for 24 h in medium (Med) or with autologous DCs generated from blood monocytes that were pretreated for 1h with culture supernatant from primary HCC cells (HCC-SN) or normal liver L02 cells (LSN) (**d–f**). Thereafter, B cells were purified by CD19[+] beads and then culture for additional 24 h in medium alone. Expression of CD83, CD86 and HLA-DR in DCs before co-culture (**a,d**) and expression of FcγRII in B cells after co-culture (**b,e**) were determined by FACS. Concentration of IL-10 in culture supernatant was determined by ELISA (**c,f**). Results represent three independent experiments (*n* = 5). (**g**) Kinetic effects of HCC-SN and LSN on Erk, p38, Jnk and IκBα activation in monocytes. Results represent three independent experiments (*n* = 4). (**h,i**) Suppression of IκBα, but not Erk, p38, or Jnk, in HCC-SN-pretreated monocytes attenuated HCC-SN-DC-mediated IL-10-producing FcγRII[low/ −] B-cell generation. (**j–l**) Pre-exposure of monocytes to HCC-SN for 1 h in the presence of a hyaluronan-specific blocking peptide (Pep-1) could partially inhibit the semimaturation of DCs, as well as subsequent generation of IL-10-producing FcγRII[low/ −] B cells. Uncropped western blot images are shown in Supplementary Fig. 3c. Results represent three independent experiments (*n* = 4). *P < 0.05, **P < 0.01 (Student's *t*-test). Error bars, s.e.m.

CD95–CD95L axis may participate in the TDC-mediated B-cell activation and IL-10 production. Accordingly, we used specific neutralizing antibody to abolish the effects of CD95L in the conditioned medium from TDCs. As expected, this treatment effectively restored B-cell FcγRII expression and suppressed IL-10 production (Fig. 4g,h). In addition, NF-κB is the downstream signaling pathway of CD95–CD95L axis[26,27], inhibiting the IκBα signalling in B cells also markedly attenuated the TDC-mediated B-cell FcγRII decrease and IL-10 production (Fig. 4i,j). These results clearly show that generation of CD95L-elicited activation is a prerequisite for the ability of TDCs to cause B-cell activation and IL-10 production.

**FcγRII[low/ −] B cells suppress T-cell function via IL-10.** Many of the B cells (76.5 ± 12.8%) were in close contact with CD8[+] T cells

in the peritumoral stroma of HCC tumours (Fig. 5a). Inasmuch as activated FcγRII[low/ −] B cells in HCC tumours potently correlated with disease progression and even produced protumorigenic IL-10 (Fig. 1d,f), we investigated whether FcγRII[low/ −] B cells affected CD8[+] cytotoxic T-cell immunity in tumours. Supporting the protumorigenic roles of FcγRII[low/ −] B cells, high infiltration of FcγRII[low/ −] B cells in HCC tumours positively correlated with dysfunction of tumour-derived CD8[+] T cells with impaired capacities for production of proinflammatory TNF-α and IFN-γ and cytotoxic granzyme B and perforin (Fig. 5b). Of note, CD8[+] T cells from HCC tumours mainly exhibited a CCR7[−] CD45RA[−] effector memory phenotype and these cells expressed higher level of IL-10 receptors (IL-10R) than the paired blood CD8[+] T cells (Fig. 5c and Supplementary Fig. 5a), which suggests that the activated IL-10-producing B cells can regulate CD8[+] T-cell function via IL-10 signals in tumours.

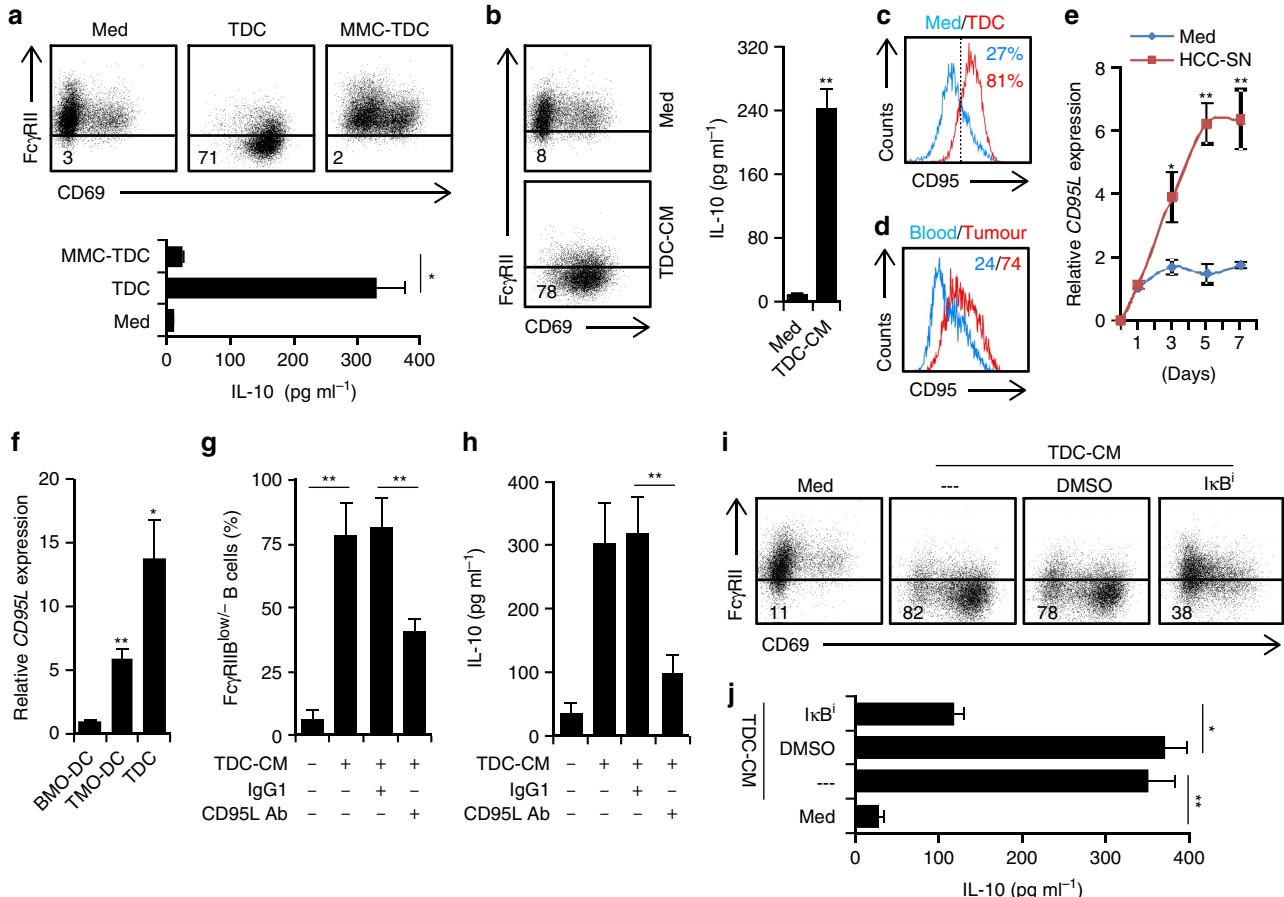

**Figure 4 | Tumour DCs induced activated FcγRII$^{low/-}$ IL-10-producing B cells through CD95–CD95L interaction.** (**a,b**) Peripheral CD19$^+$ B cells were cultured in medium or with TDCs (**a**), TDCs pretreated with mitomycin C (MMC-TDC) (**a**), or TDC-derived conditioned medium (TDC-CM) (**b**) for 24 h. Thereafter, B cells were purified by CD19$^+$ beads and then cultured for additional 24 h in medium alone. Expression of FcγRII in B cells was determined by FACS. Concentration of IL-10 in culture supernatant was determined by ELISA. (**c**) FACS analysis of CD95 in B cells cultured for 24 h in medium or with TDCs. (**d**) FACS analysis of CD95 in B cells from HCC tumour and paired blood ($n = 5$). (**e**) Kinetic expression of CD95L in DCs generated from blood monocytes that were pretreated for 1 h with HCC-SN or LSN. (**f**) Expression of CD95L in TDCs or in DCs generated from blood or tumour monocytes (BMO-DC or TMO-DC; $n = 5$). (**g,h**) Blocking CD95–CD95L interaction impaired TDC-CM-elicited IL-10-producing FcγRII$^{low/-}$ B-cell generation. (**i,j**) Inhibiting CD95 downstream IκBα activation impaired TDC-CM-elicited IL-10-producing FcγRII$^{low/-}$ B-cell generation. Results shown (**a–c,e,g–j**) represent three independent experiments ($n = 4$). *$P < 0.05$, **$P < 0.01$ (Student's $t$-test). Error bars, s.e.m.

Considering that FcγRII$^{low/-}$ B cells in mouse model lacked the ability to produce protumorigenic IL-10 (ref. 16), we therefore established an *ex vivo* system to investigate the effects of FcγRII$^{low/-}$ B cells on human tumour immunity. The FcγRII$^{low/-}$ B cells were purified from HCC tumours and then cultured directly with autologous tumour CD8$^+$ T cells. The FcγRII$^{low/-}$ B cells did induce dysfunctional CD8$^+$ T cells that exhibited impaired production of anti-tumorigenic TNF-α and IFN-γ (Fig. 5d,e). Consistent with our hypothesis, shielding the IL-10R in CD8$^+$ T cells markedly restored the ability of these cells to produce TNF-α and IFN-γ (Fig. 5d,e). Tumour FcγRII$^{low/-}$ B cells only weakly attenuated the polyclonal stimulation-mediated CD8$^+$ T-cell proliferation (Supplementary Fig. 5b). Furthermore, similar results were obtained when using FcγRII$^{low/-}$ B cells that were induced by HCC-SN-treated DCs: FcγRII$^{low/-}$ B cells suppressed the expression of proinflammatory TNF-α and IFN-γ and cytotoxic granzyme B and perforin in autologous tumour-derived CD8$^+$ T cells via an IL-10-dependent manner (Supplementary Fig. 5c). These findings show that IL-10 signals contribute to activated B cell-mediated cytotoxic T-cell suppression in tumours.

## Discussion

Although cancer patients display a widespread immunosuppressive status, there is an increased evidence that the immune activation at a tumour site can promote cancer progression[28–30]. We have previously shown that activated monocytes are enriched mainly in human hepatomatissue, where they promote disease progression by fostering pro-inflammatory response[9,29]. The present study substantially demonstrates that the FcγRII$^{low/-}$ activated B cells in a cancer environment produce protumorigenic IL-10 to suppress cytotoxic T-cell function, representing a link between immune activation and immunosuppression in the cancer environment.

The peritumoral environments in most tumours contain considerable amount of immune cells, which was previously considered as the host response to the tumour[8,31]. In the current study, we observed that B cells in the peritumoral stroma exhibited an activated phenotype with increased expression of CD69, but radically reduced FcγRII and BTLA. However, data from *ex vivo* study showed that these FcγRII$^{low/-}$ activated B cells were unable to stimulate effective antitumour T-cell responses, instead they suppressed the cytotoxic T-cell immunity, which suggests that such activated B cells can

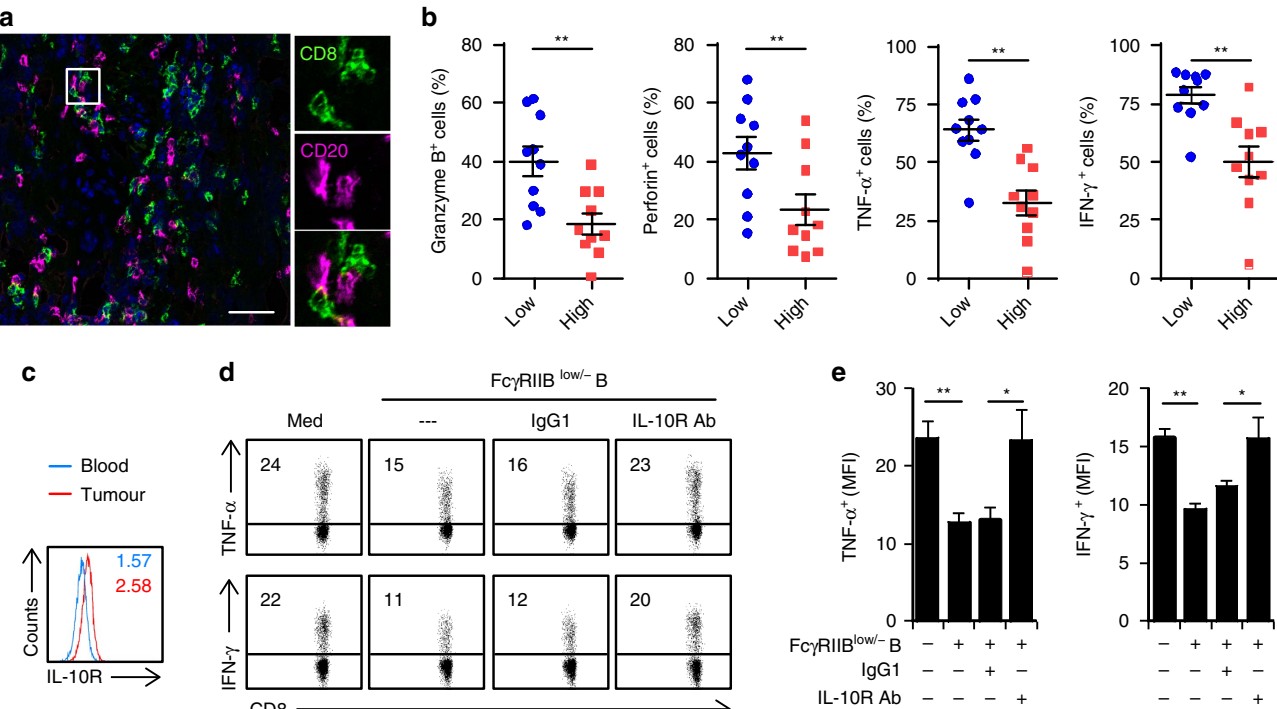

**Figure 5 | HCC-derived FcγRII^low/− B cells suppressed cytotoxic T-cell function via IL-10 signals.** (**a**) Detection of CD8 (green) and CD20 (red) (n = 6) in peritumoral stroma by immunofluorescence. Scale bar, 100 μm. (**b**) Associations of tumour FcγRII^low/− B cells with granzyme B^+, perforin^+, TNF-α^+ and IFN-γ^+ CD8 T cells (n = 20). (**c**) Expression of IL-10R on CD8^+ T cells from paired blood and HCC tumour were determined by FACS (n = 6). (**d,e**) HCC tumour-derived CD8^+ T cells were left untreated or cultured with sorted tumour FcγRII^low/− B cells in the presence or absence of an anti-IL10R or a control Ab. FACS analysis of TNF-α, IFN-γ expression in CD8^+ T cells is shown. Results represent three independent experiments (n = 5). *P < 0.05, **P < 0.01 (Student's t-test). Error bars, s.e.m.

actually benefit tumour progression. This notion is supported by our finding that the proportion of FcγRII^low/− activated B cells in tumour tissue positively correlated with advanced disease stages in HCC patients. Consistent with this, we also observed that high infiltration of FcγRII^low/− B cells in HCC tumours positively correlated with increased dysfunction of CD8^+ T cells with impaired capacities for production of proinflammatory TNF-α and IFN-γ and cytotoxic granzyme B and perforin.

B cells are polyfunctional, plastic cells that can respond to environmental factors through diverse programs[32,33]. We newly found that proinflammatory IL-17^+ cells in the peritumoral stroma stimulated hepatoma cells to produce CXCL9, CXCL10 and CXCL11, which in turn led to CXCR3^+ B-cell recruitment, maturation and IgG production[30]. In this study, we demonstrated that semimature DCs in the peritumoral stroma promoted the generation of FcγRII^low/− activated B cells. We also found that early activation of monocytes in the HCC environments was vital for semimature DC-mediated FcγRII^low/− activated B-cell generation. Several of our observations support the notion that abnormal development of monocyte-derived DCs in HCC environments contribute to B cell immunopathology. First, although high infiltration of S100^+ DCs and CD68^+ macrophages in the peritumoral stroma both predicted poor survival in HCC patients, only the density of S100^+ DCs positively correlated with the proportion of FcγRII^low/− activated B cells in HCC samples. Second, TDCs, but not the TAMs, strongly induced FcγRII^low/− activated B cells exhibiting functions to produce IL-10 and suppress cytotoxic T cells. Third, DCs differentiated from tumour-isolated monocytes (TMO-DCs) also displayed semimature phenotype as that exhibited by TDCs, and that they markedly induced B-cell activation and IL-10 production. Fourth, pre-exposure of

monocytes to primary HCC-SN could result in the formation of semimature DCs that effectively triggered B-cell activation and IL-10 production. Therefore, induction of semimature DCs from tumour-activated monocytes may represent a mechanism that reprograms B-cell activation and function.

CD95/CD95L-mediated activation-induced cell death is known to play an important role in maintenance of peripheral lymphocyte homeostasis[24,25]. However, in the current study, we identified an activation role of CD95–CD95L axis in tumours and found that blockade of CD95L effectively attenuated TDC-elicited generation of FcγRII^low/− activated IL-10-producing B cells. Indeed, other studies have shown that, on apoptosis-resistant cancer cells, stimulation of CD95 results in activation of NF-κB and MAP kinases inducing increased motility and invasiveness of cancer, which reflects the function of CD95 as a tumour promotor[34]. This finding is compatible with our current study showing that inhibiting the activation of CD95L downstream signalling pathway IκBα in B cells also markedly attenuated the TDC-mediated B-cell activation and IL-10 production. In addition, we previously demonstrated that DCs generated in cancer environments triggered rapid downregulation of CD3ε and subsequent apoptosis in autologous T cells through oxygen-dependent pathways[17]. Nevertheless, in the present culture system, DCs generated in cancer environments could not elicit marked apoptosis of autologous B cells. Therefore, it is tempting to suggest that distinguished mechanisms are employed by tumour DCs to modulate the functional status of B cells and T cells.

Studies in human peripheral blood and tumour tissue have identified CD24^high CD38^high and PD-1^high regulatory B cells, respectively, and it has been suggested that, upon encountering external stimulus, these cells regulate T-cell responses by releasing

protumorigenic IL-10[19,20,35]. In our investigation, the FcγRII[low/−] activated B cells, without additional stimulation, could secrete IL-10 to repress tumour-specific cytotoxic T-cell response. This reveals that FcγRII[low/−] activated B cells represent a regulatory B-cell subset that has not been previously defined in human cancer tissues. More interestingly, IL-10 can act as an important regulator in promoting B-cell activation, proliferation, and maturation, although its suppressive role in limiting myeloid cell and T-cell functions has been established[36,37]. It is plausible that IL-10 serving as a positive feedback during B-cell activation in cancer environments is utilized by tumours to suppress cytotoxic T-cell functions. Compatible with this, IL-4 is a regulator for suppressive M2 macrophages polarization, but it effectively induces B-cell activation and maturation[38].

In mouse models, FcγRII[high] B cells have been defined as an IL-10-producing regulatory B-cell subset[16]. However, in human HCC tumours, the FcγRII[low/−], but not the FcγRII[high], B cells actively and directly produced IL-10, suggesting the immune cellular networks between humans and mice are very different. In fact, several recent studies also demonstrated existences of the TGF-β[+] and B7-H1[+] regulatory B cells in mouse models[39,40]. However, we have found that, in human HCC tumours, B cells are unable to express TGF-β and B7-H1 (unpublished data). Thus, studying the composition and functions of infiltrating B cells in human tumours may help us better understand the actual roles of B cells in tumour pathogenesis.

While most studies focus on T cells in tumour-infiltrating lymphocytes in human tissues[41–43], the current study performed a comprehensive study on FcγRII[low/−] activated regulatory B cells in human HCC tumours. The current study provides important insights into possible manipulation of B-cell activation-mediated immunosuppression in human tumours. Soluble factors released by tumour DCs promote the generation of FcγRII[low/−] activated B cells by triggering CD95. Thereafter, the FcγRII[low/−] activated B cells operate via IL-10-dependent pathways to induce T-cell suppression and thereby create circumstances that are favourable to tumour growth. Accordingly, immunotherapies that interfere with CD95–CD95L axis and IL-10 production should be part of the arsenal used to re-establish immune response in cancer patients. Thus selectively modulating functional activities of activated stroma cells in patients may provide strategies for anticancer immunotherapies[44–48].

## Methods

**Patients and specimens.** Liver and tumour tissue samples were obtained from HCC patients during surgery at the Cancer Center of Sun Yat-sen University (Supplementary Table 1). None of the patients had received anticancer therapy before the sampling, and individuals with concurrent autoimmune disease, HIV, or syphilis were excluded. Paired samples of blood and tumour tissue with invading edge from 25 HCC patients who received treatment between 2013 and 2016 (cohort 1; Supplementary Table 1) were used to isolate peripheral and tumour-infiltrating leukocytes. An additional 135 HCC patients who had undergone curative resection between 2007 and 2011 and had complete follow-up data (cohort 2; Supplementary Table 1) were enrolled for analysis of survival. Samples of normal liver tissue were obtained distal to liver hemangiomas (n = 2). Clinical stages were classified according to the guidelines of the International Union against Cancer. All samples were anonymously coded in accordance with local ethical guidelines (as stipulated by the Declaration of Helsinki), and written informed consent was obtained and the protocol was approved by the Review Board of Sun Yat-sen University Cancer Center.

**Immunostaining of tissue sections and cytocentrifuged cells.** Paraffin-embedded formalin-fixed HCC samples were cut into 5-μm sections, which were processed for immunohistochemistry[49]. The sections were incubated with Abs against human CD20 (ZSBio), S100 (ZSBio) and/or CD68 (Dako), and then stained in an Envision System (DakoCytomation). For immunochemistry analysis, cytocentrifuged CD11c[high]CD11b[high] DCs and CD11c[low]CD11b[high] macrophages were fixed and incubated with Ab against human S100, and then stained in an Envision System (DakoCytomation). For immunofluorescence analysis, frozen

sections were stained with mouse anti-human CD8 (Neomarkers) plus rabbit anti-human CD20, followed by Alexa Fluor 488-conjugated anti-mouse IgG plus Alexa Fluor 555-conjugated anti-rabbit IgG (Molecular Probes). Positive cells were quantified using ImagePro Plus software or detected by confocal microscopy.

**Evaluation of immunohistochemical variables.** Analysis was performed by two independent observers who were blinded to the clinical outcome. At low-power field (×100), the tissue sections were screened; and the five most representative fields were selected using a Leica DM IRB inverted research microscope (Leica Microsystems, Wetzlar, Germany). For evaluating the density of tissue infiltrating S100[+] cells, the respective areas of nontumoral liver, peritumoral stroma, and intratumoral region were then scanned at ×400 magnification (0.146 mm[2] per field). The number of nucleated S100[+] cells in each area was then counted manually and expressed as cells per field. Positively stained cells that are smaller than the size of circulating T cells were excluded from counting. The average counting by two investigators was applied in the following analysis to minimize interobserver variability.

**Preparation of HCC-SN and LSN.** Culture supernatants were acquired by culture of completely digested HCC tumour biopsy specimens. All specimens were from individuals without concurrent autoimmune disease, HBV, HCV, HIV or syphilis. The digested tumour or liver cells were washed in medium containing polymyxin B (20 μg ml[−1], Sigma-Aldrich) to exclude endotoxin contamination. Thereafter, 10[7] digested cells were resuspended in 10 ml of complete medium and cultured in 100-mm dishes. After 2 days, the supernatants were harvested, centrifuged, and stored at −80 °C.

The human normal liver cell line L02 were obtained in January 2013 from the Cell Bank of Type Culture Collection of the Chinese Academy of Sciences (Shanghai, China). The cells were authenticated by short tandem repeat profiling and were confirmed to be mycoplasma-negative before use, and they were maintained in RPMI1640 supplemented with 10% FCS. Culture Supernatant was prepared as that shown in our previous study[17]. In detail, 5 × 10[6] tumour cells were plated in 10 ml of complete medium in 100-mm dishes for 3–4 days, and the supernatants were subsequently centrifuged, filtered and stored in aliquots at −80 °C.

**Isolation of mononuclear cells from blood and tissues.** Peripheral mononuclear leukocytes were isolated by Ficoll density gradient centrifugation, and fresh tissue-infiltrating mononuclear leukocytes were obtained as that shown in our previous study[28]. In short, liver biopsy specimens were cut into small pieces and digested in RPMI 1640 supplemented with 0.05% collagenase IV, 0.002% DNase , and 20% FBS at 37 °C for 1 h. Dissociated cells were filtered through a 150-mm mesh and separated by Ficoll centrifugation, and the mononuclear cells were washed and resuspended in RPMI 1640 supplemented with 10% FBS. The whole isolation process or digestion buffer (containing 20% FBS) would not affect the survival of CD45[+] mononuclear cells. Purification of B cells, T cells or monocytes from the leukocytes was achieved with a MACS column purification system (Miltenyi Biotec). The FcγRII[low/−] or FcγRII[high] tumour B cells, CD24[high]CD38[high] blood B cells, CD45[+]CD15[−]CD11c[high]CD11b[high] TDCs, as well as CD45[+]CD15[−]CD11c[low]CD11b[high] TAMs were further sorted using FACS (Moflo, Beckman Coulter).

***In vitro* generation of tumour-related DCs.** The protocol for *in vitro* generation of tumour-related DCs was established and modified according to that shown in our previous study[17]. In short, to generate TMO-DCs, BMO-DCs, HCC-SN/DCs and LSN/DCs, tumour-derived CD14[+] monocytes, blood monocytes, and blood monocytes that were pretreated with 30% culture supernatant from primary HCC cells (HCC-SN) or L02 cells (LSN) for 1 h were cultured for 6 days in complete RPMI medium supplemented with 40 ng ml[−1] GM-CSF and IL-4 in the presence or absence of a specific inhibitor of the Jnk (SP 600125, 5 μM), Erk (U0126, 20 μM), NF-κB (BAY 11-7082, 5 μM), or p38 (SB 203580, 20 μM) signal. Half of the culture medium was replaced on day 3. BMO-DC maturation was induced by stimulating the cells with 200 ng ml[−1] LPS for 24 h. In some experiments, the HCC-SN was pre-incubated with a hyaluronan-specific blocking peptide (Pep-1, 200 μg ml[−1]) or control peptide (Cpep) before exposure to monocytes.

**Flow cytometry.** B cells, T cells, monocytes, macrophages and DCs from peripheral blood, tissues, or *in vitro* culture were stained with fluorochrome-conjugated antibodies and then analysed by flow cytometry (FACS). Tumour-infiltrating lymphocytes and T cells from *in vitro* culture were stimulated at 37 °C for 5 h with Leukocyte Activation Cocktail (BD Pharmingen). Thereafter, cells were stained with surface markers, fixed and permeabilized with IntraPrep reagent (Beckman Coulter), and finally stained with intracellular markers. Data were acquired on a Gallios flow cytometer (Beckman Coulter) and analysed with FlowJo software (6.0.0.0). The fluorochrome-conjugated antibodies used are listed in Supplementary Table 3.

**Real-time PCR.** Trizol reagent (Invitrogen) was used to isolate total RNA of B cells, monocytes, or DCs from HCC tumour and paired blood or from an *in vitro* culture system. Aliquots (2 μg) of the RNA were reverse-transcribed using MMLV reverse transcriptase (Promega). The specific primers used to amplify the genes are listed in Supplementary Table 4. The PCR was performed in triplicate using SYBR Green Real-Time PCR MasterMix (TOYOBO) in a Roche LightCycler 480 System. All results are presented in arbitrary units relative to 18S rRNA expression.

**ELISPOT assay.** ELISpot assays were performed using commercial sets (BD Pharmingen) according to the manufacturer's instructions. For IL-10 detection assay, $1 \times 10^5$ purified tumour B cells, FcγRII$^{low/-}$ B cells and FcγRII$^{high}$ B cells were cultured for 48 h. In other case, $1 \times 10^5$ sorted blood CD38$^{high}$CD24$^{high}$ B cells were left untreated or stimulated with 1 μg ml$^{-1}$ CD40L plus 2 μg ml$^{-1}$ CpG for 24 h. The images were scanned with ELISpot Reader (CTL) and spot numbers were counted manually.

**Enzyme-linked immunosorbent assay.** Concentrations of the inflammatory cytokines IL-10, IFN-β1, IFN-β2 and immunoglobulins in the supernatants from *in vitro* culture systems were detected using enzyme-linked immunosorbent assay (ELISA) kits according to the manufacturers' instructions (eBioscience for cytokines and Bethyl Laboratories, Inc. for immunoglobulins).

**Immunoblotting.** The cells were washed three times with PBS and the pellets were resupended in lysis buffer for 20 min on ice. After centrifugation at 10,000$g$ for 10 min, the supernatants were dissolved in Laemmli sample buffer and heated at 95 °C for 5 min. Equal amount of cellular proteins were separated on 10% SDS–polyacrylamide gel electrophoresis and electrotransfered to nitrocellulose membranes. The membranes were blocked with 3% bovine serum albumin, and the presence of indicated protein on the blots was detected with specific Ab and a commercial ECL kit[50]. The antibodies and their dilutions are listed in Supplementary Table 5.

**Ex vivo B-cell culture system.** Peripheral B cells were left untreated or cultured for indicated time with TDCs, TAMs, DCs generated from HCC blood- or tumour-derived monocytes (BMO-DCs or TMO-DCs), DCs generated from normal blood monocytes that had been pretreated for 1 h with HCC-SN or LSN in the presence or absence of a specific inhibitor of the Jnk, Erk, NF-κB, or p38 signal at a ratio of 20:1. In other experiments, peripheral B cells were cultured with TDCs that were pre-treated for 20 min with mitomycin C (10 μg ml$^{-1}$) or were cultured in conditioned medium from TDCs (TDC-CM) in the presence or absence of IFN-β1, IFN-β1, CD40L or CD95L Ab, a control Ab or a specific inhibitor of NF-κB (BAY 11–7082, 5 μM). Thereafter, B cell FcγRII expression and IL-10 production were determined by FACS and ELISA, respectively.

**Ex vivo plasma cell induction.** Tumour B-cell subsets sorted by FACS or blood B cells left untreated or cultured with TDCs were incubated in RPMI-1640 supplemented with 10% FBS and 1 μg ml$^{-1}$ CD40L (R&D Systems) plus 50 ng ml$^{-1}$ recombinant IL-21 (PeproTech). Culture supernatants were collected on Day 6 for detection of immunoglobulins.

**Ex vivo tumour T-cell immunosuppression.** HCC tumour-derived CD8$^+$ T cells were left untreated or cultured at a ratio of 2:1 with autologous FcγRII$^{low/-}$ B cells isolated from tumour tissue or induced by TDCs in the presence or absence of an IL-10R Ab or a control Ab for 48 h. Thereafter, the expression of proinflammatory TNF-α and IFN-γ and cytotoxic granzyme B and perforin in CD8$^+$ T cells were examined by FACS. In other case, the CD8$^+$ T cells cultured at a ratio of 2:1 with autologous FcγRII$^{low/-}$ B cells were stimulated with 2 μg ml$^{-1}$ anti-CD3 plus 2 μg ml$^{-1}$ anti-CD28 for 4 days, the proliferation of T cells were determined by staining of Ki67.

**In vivo mouse experiment.** Murine Hepa1-6 or H22 hepatoma cells ($10^6$) in 25 μl Matrigel (Corning) were injected under the hepatic capsule of 5–7-week-old female C57BL/6 mice[51]. After 25 days, the peripheral and tumour leukocytes were isolated, stained with a fluorochrome-conjugated mAb, and analysed by FACS. All mice were randomly grouped, and observers were blinded to the classification. The animal use protocols were reviewed and approved by the Institutional Animal Care and Use Committee of Sun Yat-sen University.

**Statistical analysis.** Results are expressed as means ± s.e.m. Group data were analysed by ANOVA followed by Student's *t*-test or log-rank test. Relationships between parameters were assessed by Pearson correlation analysis and linear regression analysis. All data were analysed using two-tailed tests unless otherwise specified, and $P < 0.05$ was considered statistically significant. No statistical method was used to predetermine sample size.

**Data availability.** All the relevant data are available from the authors upon request.

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

## Acknowledgements

This work was supported by project grants from the National Key Research and Development Plan of China (2016YFA0502600), the National Natural Science Foundation of China (81422036, 31470855 and J1310025), the Guangdong Natural Science Funds for Distinguished Young Scholar (S2013050014639), the Foundation for the Author of National Excellent Doctoral Dissertation of PR China (201230), the Project Supported by Guangdong Province Higher Vocational Colleges & Schools Pearl River Scholar Funded Scheme (2016), and the Fundamental Research Funds for the Central Universities (15lgjc09).

## Author contributions

F.-Z.O. performed most of the experiments, analysed the results; R.-Q.W. and Y.W. contributed to FACS and analysed the data; D.Y., R.-X.L. and X.X. did immunohistochemical staining and image analysis; L.Z and B.L. made technical and intellectual contributions; X.-M.L. and D.-M.K. contributed to study design, supervised the study, and contributed to writing the manuscript.

## Additional information

**Competing financial interests:** The authors declare no competing financial interests.

