## [Peer Review File · Nature Communications]

Reviewers' comments:

Reviewer #1 : Expert in Tumor immunology

(Remarks to the Author):

The study reports characterization of a new subset of B cells that exhibit potentially functions in humans HCC. These B cells express low levels of FcγRII highly produce IL10 and suppress effector CD8+T cells. Although these cells are not to found in circulation, their presence in peritumoral stroma positively correlates with the stage of HCC. Interestingly, tumor-infiltrating semimature DCs, which is also co-localized with B cells and positively associated with a disease outcome, induce these B cells. The authors show that tumor-DCs induce B cells using CD95/CD95L axis. Overall, this is an interesting paper that presents a new insight with a potential clinical implication. The presentation is concise and clear, although some experimental procedures need to be better described. The results seem to be interpreted correctly, although I do find a few minor issues (as follows).

1. The conclusion that only TDCs induce IL-10 from FcγRII-low B cells appears to be misleading, since B cells treated with healthy donor iDCs and mDCs also up regulated IL-10 (SFig.2C). Thus, please correct the statement that these two types of DCs do not activate IL-10 production in B cells (1st paragraph, p.7).

2. I am quite puzzled that 1h pretreatment was sufficient to differentiate CD14+ MO into DCs. There is no proof shown that MO were differentiated into DCs.

Thus, why the authors call them DCs - do they express DC phenotype? Shouldn't these cells be called monocytes? Can untreated monocytes also induce B cells to express IL-10 and down regulate FcγRII?

3. The results on CD95/CD95L axis are interesting. However, the viability of B cells upon TDC treatment is not known. Can the authors exclude a simple possibility that TDCs may only induce

a preferential survival of FcγRII-low B cells (as they may express less CD95L or CD95)? Thus, is it "induction" or preferential survival of B cells?

4. Since mitomycin C can be cytotoxic for DCs, the treated DCs could be dead and thus failed to down regulate FcγRII/up regulate IL-10 in B cells shown in Fig.4a.

5. The experimental procedures are poorly described. At least, methods for TMO-DC generation and cytomycin C use need to be presented.

6. Please indicate the source and the assay used for CD8+T cells in Fig.5B. Are these cells TILs and what were stimuli used prior to staining?

Reviewer #2: Expert in HCC microenvironment

(Remarks to the Author):

In this study, Ouyang et al. reported a population of tumor B cells expressing low FCγRII in human HCC. They first showed that this subset of tumor B cells secreted IL-10, a well-known immune suppressive cytokine. By evaluating the FCγRII expression and IL10 secretion, they went on to show that activation of FCγRIIlow tumor B cells was mediated through tumor associated dendritic cells (TDCs) but not tumor associated macrophages (TAMs). They showed that TDCs were derived from monocytes which were educated by the HCC tumors. Specifically, they showed that the HCC tumors activated the IκBα pathway in the monocytes to initiate the TDC-B cell activation. Next, they provided evidence that FCγRIIlow tumor B cells were not responsive to conventional stimulant of B regulatory cells, CD40L, suggesting that this is a new population of B regulatory cells with distinct stimulatory mechanisms. They went on to show that the TDCs activated the FCγRIIlow tumor B cells through CD95L/CD95 axis, subsequently leading to the production of IL10 and inhibition of CD8 T cells. Blocking CD8 T cells reversely restored activity of CD8 T cells. Clinically, the authors showed that FCγRIIlow tumor B cells correlated with more advanced HCC stages and impaired CD8 T cells in HCC. In general, the authors investigated a complicated and interesting mechanism by which tumors educate the immune cells in the microenvironment and showed the interplay of different immune populations in HCC. They have also identified a unique subset of B regulatory cells in HCC. The story is

interesting and the manuscript is clearly written, but the study overall is lack of depth. There are a number of key missing links in the study.

More specific comments:

1. The authors showed that the supernatant from primary HCC was able to activate the monocytes. However, the responsible components from HCC have not been studied or clarified. This is the most important and interesting piece of information. Elucidation of the key components in HCC cells that initiate the B reg activation is essential to the development of new drug targets.

2. There are many pieces of interesting information in the study; however, the authors did not provide solid evidence to link the information together. For example, the authors independently showed that (1) HCC activated monocytes, (2) monocytes differentiated into TDC, (3) TDC activated B regulatory cells, (4) B regulatory cells inactivated CD8 T cells. Nonetheless, the authors did not show comprehensively how these mechanisms are interlinked as a system. This is greatly limited by the lack of animal model in the study. The authors briefly mentioned that the human and mouse immune networks are different and the FC γ RIII^{low} B cells in mice do not produce IL-10. However, whether this is a general phenomenon in mice (have different strains been tested?) and whether HCC bearing mice have enrichment of IL10-producing FC γ RIII^{low} B cells have not been clearly addressed. Although it is an advantage to perform ex-vivo experiments with the immune cells and HCC cells from patients, an animal model (knockout mice) is indispensable to consolidate the hypothesis.

3. The same research group recently reported another population of IL10 producing regulatory B cells in HCC which express CD5^{hi}CD24^{-/+}CD27^{hi/+}CD38^{dim}PD1^{hi} (Xiao et al. Cancer Discovery. 2016). However, the authors did not explain the difference between this population and the FC γ RIII^{low} B cell population. Is there any overlapping between these two populations? What are their functional differences? Which population is the dominant form? If they both exist, what are their functional differences? Or, if they cooperate with each other? Experimental evidence is needed.

4. The authors did not provide clear and strong rationales for some experiments. For example, why the authors only studied the p38/JNK/ERK and I κ B pathways in monocytes. Are there any other pathways that control the differentiation of monocytes? It is not clear why the authors jumped from studying IFN- β 1/ β 2 to CD95L in TDCs.

Reviewer #3 : Expert in Tumor immunology

(Remarks to the Author):

The manuscript entitled "Dendritic cell-elicited B-cell activation fosters immune privilege via IL-10 signals in hepatocellular carcinoma" by Ouyang et al highlights the contribution of B cells to the immunosuppressive tumor microenvironment. They are showing the specific induction of a FcγRIIIlow/- B cells that inhibit cytotoxic T cell responses in an IL-10 dependent manner. Although these findings are very relevant and pose potential therapeutic targets for cancer treatment, there are some concerns.

The most important concern is the lack of mentioning/describing/testing of the different isoforms of FcγRII. In humans, at least two isoforms of FcγRII are present. FcγRIIa has been described as a potent immune-activating receptor and it contains an ITAM motif, capable of mediating phagocytosis, ADCC, and initiation of inflammatory cytokine release. FcγRIIb has an ITIM motif and is an immune-inhibiting receptor. Both isoforms have been shown to be present on human B cells. Unfortunately, the monoclonal antibody used 6C4, specifically recognizes the two isoforms of human FcγRII. Therefore, the contribution of either one of the isoforms can not be distinguished nor commented on.

Major comments:

- The authors recently published another novel regulatory B cell population that promotes disease progression in hepatoma. This population expressed high levels of PD-1. Is this the same population? Are the FcγRIIIlow/- B cells also expressing PD-1?
- The way DCs are defined in the manuscript is not precise. With immunohistochemistry DCs are simply defined as S100+, which may include a wide variety of cells. The expression of some S100 molecules has been associated with myeloid-derived suppressor cells (MDSCs), which are enriched in the tumor microenvironment. Further on, when these tumor DCs are sorted as CD45+CD15-CD11chiCD11bhi cells, a definition that encompasses a lot of myeloid populations, including monocytes and MDSCs. The authors should be very careful how they call the cells they are using or simply use more markers to pinpoint the populations they are using. An example of the gating strategy should also be shown.
- The authors are claiming that monocytes within the local tumor environment are transformed into semimature DCs. The authors have not mentioned anywhere in the manuscript how they are generating these monocyte-derived DCs? Is it with the classical protocol of ex vivo treatment with IL-4 and GM-CSF or are they spontaneously differentiating into DCs? If it is the first situation, making the claim that monocytes are differentiating into semimature DCs within the tumor environment is not at all correct, unless the authors prove that the tumor microenvironment contains IL-4 and GM-CSF. A very relevant question that comes here is what kind of effects does the tumor environment have on monocytes (without the additional treatment with IL-4 and GM-CSF) and most importantly on primary DCs (and not monocyte-derived DCs).

- The way HCC supernatants are prepared doesn't preclude immune cell-derived cytokines and factors. If the pure effect of HCC cells is sought after, then these cells should be purified prior to supernatant preparation. As mentioned above, the effect of such supernatants on both pure monocytes and pure primary DCs should be determined.
- The semimature status of tumor-derived "DCs" might be the reason why B cells are activated and induced to produce IL-10. It is very important to compare those tumor-derived "DCs" with primary activated DCs (activated by example a TLR ligand). It is also relevant to determine the cytokine production profile of those tumor-derived "DCs" as such cytokines have major role in promoting B cell activation and antibody production and antibody class-switching.

Minor comments:

1. Figure 1b: how does FcγRII expression by B cells in healthy donors compare to that of HCC patients (both blood and healthy liver tissue). Moreover, does the same FcγRII expression pattern in HCC patients hold in other tissues (for example lymph nodes)?
2. Do FcγRII+ B cells adopt the same profile as FcγRIIIlow/- B cells when they are activated?
3. Figure 1e: what is the CD40 expression level by FcγRIIIlow/- B cells? the reason for no antibody production might be due to lack of proper stimulation.
4. Figure 2c: the authors haven't defined what is considered as "high" S100+DC density and what is "low" S100+DC density.

Altogether, it is an interesting study, but the aforementioned concerns should be addressed and more care should be taken when drawing conclusions.

We would like take this opportunity to thank the reviewers for their thoughtful and constructive comments that helped us to improve our manuscript. Considering that the journal *Nature Communications* will publish the reviewer comments to the authors and author rebuttal letters, we showed some of the new added data that supported our main conclusion only in the rebuttal letters.

Reply to Reviewer 1

1. The conclusion that only TDCs induce IL-10 from FcγRII^{low} B cells appears to be misleading, since B cells treated with healthy donor iDCs and mDCs also up regulated IL-10 (SFig.2C). Thus, please correct the statement that these two types of DCs do not activate IL-10 production in B cells (1st paragraph, p.7).

Reply: We apologize for the unsuitable description and have reformulated the statement as following: "Notably, we found that immature DCs or mature DCs generated from healthy blood monocytes only weakly affected B cell activation and subsequent IL-10 production (Supplementary Fig. 2c,d)"(**Page 7, Line 9-11**).

2. I am quite puzzled that 1h pretreatment was sufficient to differentiate CD14⁺ MO into DCs. There is no proof shown that MO were differentiated into DCs. Thus, why the authors call them DCs - do they express DC phenotype? Shouldn't these cells be called monocytes? Can untreated monocytes also induce B cells to express IL-10 and down regulate FcγRII?

Reply: We apologize for the lack of methods about the generation of tumor-related DCs. The protocol for tumor-related DCs was established and modified according to that shown in our previous study (Kuang et al, **J Immunol**, 2008, 181:3089). In this study, to generate TMO-DCs, BMO-DCs, HCC-SN/DCs, and LSN/DCs, tumor-derived CD14⁺ monocytes, blood monocytes, and blood monocytes that were pretreated with 30% culture supernatant from primary HCC cells (HCC-SN) or L02 cells (LSN) for 1 hour were cultured for 6 days in complete RPMI medium supplemented with 40 ng/ml GM-CSF and IL-4 in the presence or absence of a specific inhibitor of the Jnk (SP 600125, 5 μM), Erk (U0126, 20 μM), NF-κB (BAY 11-7082, 5 μM), or p38 (SB 203580, 20 μM) signal. Half of the culture medium was replaced on day 3. BMO-DC maturation was induced by stimulating the cells with 200 ng/ml LPS for 24 h. We have added this information into Methods section of the manuscript (**Page 18, Line 20-25; Page 19, Line 1-6**).

As shown in Fig. 3a,d in previous version of manuscript, TMO-DCs and HCC-SN/DCs, but not BMO-DCs or LSN/DCs, exhibited a semimature DC phenotype as that displayed by DCs directly isolated from HCC tumors (Fig. 2d). As suggested by the Reviewer, we cultured

healthy blood monocytes with autologous blood B cells for 3 days and did not detect FcγRII down-regulation and IL-10 production in B cells (**Fig. 1 for Reviewer**).

Fig. 1 for Reviewer: Effect of healthy blood monocytes on B cell activation and IL-10 production.

(a,b) Healthy blood B cells were cultured with autologous blood monocytes for 3 days and the expression of CD69 and FcγRII on B cells was determined by FACS. Thereafter, the production of IL-10 in sorted B cells from coculture was determined by ELISpot. Results represent three independent experiments (n = 5).

3. The results on CD95/CD95L axis are interesting. However, the viability of B cells upon TDC treatment is not known. Can the authors exclude a simple possibility that TDCs may only induce a preferential survival of FcγRII^{low} B cells (as they may express less CD95L or CD95)? Thus, is it "induction" or preferential survival of B cells?

Reply: The Reviewer brings up an interesting issue about TDC-mediated selective survival of FcγRII^{low/-} B cells. However, in the coculture system of TDCs and B cells, we did not detect a marked apoptosis of total B cells, as assessed by determining the number of dead trypan blue positive cells. We have added this information in the revised manuscript (**Page 7, Line 1-3**). In addition, we also found that, besides down-regulating FcγRII, TDCs also up-regulated the activation marker CD69 and down-regulated BTLA in B cells (Fig. 2e and **New Supplementary Fig. 2b**). These data, together with result that FcγRII^{low/-} B cells induced by TDCs expressed more CD95 (Fig. 4c), indicate that TDCs induce FcγRII^{low/-} B cells but not selectively promote the survival of those cells.

4. Since mitomycin C can be cytotoxic for DCs, the treated DCs could be dead and thus failed to down regulate FcγRII/up regulate IL-10 in B cells shown in Fig. 4a.

Reply: We agree with the Reviewer that mitomycin C can be cytotoxic for DCs. In this study, we only used a related low concentration of mitomycin C (10 μg/ml) to pretreat TDCs for 20 min. At such a concentration, mitomycin C weakly increased cell death of TDCs but

completely abolished cytokine production by those cells (**New Supplementary Fig. 4a, Page 9, Line 1-3**).

Moreover, we have shown that conditioned medium from TDCs alone could promote B cell activation and IL-10 production (Fig. 4b). Therefore, we concluded that soluble mediators released by TDCs triggered B cell activation and IL-10 production.

5. The experimental procedures are poorly described. At least, methods for TMO-DC generation and mitomycin C use need to be presented.

Reply: We thank the Reviewer for the thoughtful suggestions and have included detail description of methods for tumor-related DC generation and mitomycin C treatment in the revised manuscript (**Page 18, Line 20-25; Page 19, Line 1-6; New Supplementary Fig. 4a; Page 9, Line 1-3**).

6. Please indicate the source and the assay used for CD8⁺ T cells in Fig. 5B. Are these cells TILs and what were stimuli used prior to staining?

Reply: The CD8⁺ T cells in Fig. 5b were isolated from human HCC tumor tissues. The tumor-derived lymphocytes were stimulated at 37°C for 5 h with Leukocyte Activation Cocktail (BD Pharmingen, San Diego, CA). Thereafter, cells were stained with surface markers, fixed and permeabilized with IntraPre reagent (Beckman Coulter, Fullerton, CA), and finally stained with intracellular markers. We thank the Reviewer for the thoughtful suggestions and have emphasized this information in the revised manuscript (**Page 10, Line 22-25; Page 19, Line 11-16**).

Reply to Reviewer 2

1. The authors showed that the supernatant from primary HCC was able to activate the monocytes. However, the responsible components from HCC have not been studied or clarified. This is the most important and interesting piece of information. Elucidation of the key components in HCC cells that initiate the Breg activation is essential to the development of new drug targets.

Reply: In a previous study, we found that hyaluronan fragments (HA) constitute a common factor produced by human tumors, including hepatoma, to induce formation of suppressive macrophages through transient early activation of monocytes. Purified HA can mimic the kinetic effect of tumor culture supernatant in inducing monocyte activation. Pretreatment with anti-CD44 mAb or Pep-1, to antagonize the interactions between HA and its receptors, markedly inhibited the tumor culture supernatant- or HA-mediated monocyte activation. And, silencing of hyaluronan synthase 2 (HAS2) in tumor cells reduced the levels of HA in the tumor culture supernatant and partially blocked the induction of monocyte activation (Fig. 4B, 5B, 6D and Fig. S3 in Kuang et al, **Blood**, 2007, 110:587).

In the current study, we showed that such early activation of monocytes in hepatoma environments might contribute to the generation of semimature DCs, which in turn induced activated FcγRII^{low/-} IL-10-producing B cells. As pointed out by the Reviewer, we noted that these inhibitors could partially block the tumor culture supernatant-mediated monocyte activation (previous study) as well as the HCC-SN-mediated DC semimaturation and DC-elicited FcγRII^{low/-} IL-10-producing B cell generation (**New Fig. 3j-l in the present study**). This information has been included in the revised manuscript (**Page 8, Line 17-20**).

2. There are many pieces of interesting information in the study; however, the authors did not provide solid evidence to link the information together. For example, the authors independently showed that (1) HCC activated monocytes, (2) monocytes differentiated into TDC, (3) TDC activated B regulatory cells, (4) B regulatory cells inactivated CD8 T cells. Nonetheless, the authors did not show comprehensively how these mechanisms are interlinked as a system. This is greatly limited by the lack of animal model in the study. The authors briefly mentioned that the human and mouse immune networks are different and the FcγRII^{low} B cells in mice do not produce IL-10. However, whether this is a general phenomenon in mice (have different strains been tested?) and whether HCC-bearing mice have enrichment of IL10-producing FcγRII^{low} B cells have not been clearly addressed. Although it is an advantage to perform ex-vivo experiments with the immune cells and HCC cells from patients, an animal model (knockout mice) is indispensable to consolidate the hypothesis.

Reply: We agree with the Reviewer that it would be nice to establish a mouse hepatoma model to integrate our ex vivo findings in human HCC tumors. As suggested by the Reviewer,

we set out to establish two in vivo mouse hepatoma models. That is why the revision of this manuscript takes such a long time. However, in both Hepa1-6 hepatoma and H22-associated ascitic hepatoma models, we did not observe a marked increase of FcγRII^{low/-} B cells in hepatoma tissues compared with paired blood samples. More strikingly, very few B cells from hepatoma tissues exhibited an FcγRII^{low/-} phenotype (**Fig. 2 for Reviewer**). These data suggest that mice may do not completely reproduce the patterns of gene expression and cellular networks induced by human disease. Consistent with this, we recently showed that the absolute numbers of PD-1^{hi} B cells in mouse hepatoma tissues were not as pronounced as those in human HCC tissues (Supplementary Fig. S7C in Xiao et al, **Cancer Discov**, 2016, 6,546).

Fig. 2 for Reviewer: Expression of FcγRII on blood and tumor B cells from Hepa1-6 and H22 mouse hepatoma models.

Murine Hepa1-6 and H22 hepatoma cells (10^6) in 25 μ l Matrigel (Corning) were injected under the hepatic capsule of 5–7-week-old female C57BL/6 mice and BALB/c mice, respectively, for 25 days. Thereafter, the expression of FcγRII on blood and tumor B cells from both models was determined by FACS (n = 5 for each).

In the present investigation, to well-reflect the real functions of immune cells in human HCC tissues, we conducted most of experiments independently using immune cells directly isolated from human HCC tissues. We agree with the Reviewer that it is important to interlink these findings in a system. In our previous in vitro system, we have demonstrated that primary HCC-SN could induce the early activation of monocytes (Fig. 3g), which in turn led to formation of semimature DCs (Fig. 3d). These HCC-SN-educated semimature DCs subsequently promoted the generation of activated FcγRII^{low/-} IL-10-producing B cells (Fig. 3e,f). In the revised manuscript, we further demonstrated that the FcγRII^{low/-} B cells generated in such a system also suppressed the function of autologous CD8⁺ T cells activated by polyclonal stimulations (2.5 μ g/ml anti-CD3 and -CD28) (**New Supplementary Fig. 5c**).

Thus, the data acquired in such an in vitro system mimic all our ex vivo findings in human HCC tumors.

3. The same research group recently reported another population of IL-10 producing regulatory B cells in HCC which express $CD5^{hi}CD24^{-/+}CD27^{hi/+}CD38^{dim}PD1^{hi}$ (Xiao et al. Cancer Discovery. 2016). However, the authors did not explain the difference between this population and the $Fc\gamma RII^{low}$ B cell population. Is there any overlapping between these two populations? What are their functional differences? Which population is the dominant form? If they both exist, what are their functional differences? Or, if they cooperate with each other? Experimental evidence is needed.

Reply: The issue raised by the Reviewer is very attractive. We accordingly analyzed the expression of $Fc\gamma RII$ and PD-1 in HCC-infiltrating B cells during this round of revision. As shown in **Fig. 3 for Reviewer**, ~60% PD-1^{hi} B cells exhibited an $Fc\gamma RII^{low/-}$ phenotype, whereas < 10% $Fc\gamma RII^{low/-}$ B cells were PD-1^{hi}. Interestingly, the sorted PD-1^{hi} $Fc\gamma RII^{low/-}$ B cells from HCC tissues, without PD-1 triggering, could not produce IL-10. By contrast, the PD-1^{low/-} $Fc\gamma RII^{low/-}$ B cells, without additional stimulation, were the major source of IL-10 production in tumor B cells. Therefore, over 90% $Fc\gamma RII^{low/-}$ B cells were not PD-1^{hi} B cells, and they acquired ability to produce IL-10 via a mechanism differed from PD-1^{hi} B cells.

Fig. 3 for Reviewer: Analysis of PD-1 and $Fc\gamma RII$ expression, as well as IL-10 production, in B cells derived from human HCC tumor tissues.

(a) Analysis of PD-1 and $Fc\gamma RII$ expression on B cells derived from human HCC tumor tissues. (b) Sorted PD-1^{hi} $Fc\gamma RII^{low/-}$ B cells or PD-1^{low/-} $Fc\gamma RII^{low/-}$ B cells were left untreated or stimulated with PD-1 agonist or control goat IgG for 24 hr. The production of IL-10 was determined by ELISpot. Results represent three independent experiments (n = 5).

4. The authors did not provide clear and strong rationales for some experiments. For example, why the authors only studied the p38/JNK/ERK and I κ B pathways in monocytes? Are there any other pathways that control the differentiation of monocytes? It is not clear why the authors jumped from studying IFN- β 1/ β 2 to CD95L in TDCs.

Reply: We apologize for the incomplete description. Inasmuch as both p38/JNK/ERK and I κ B pathways have been considered as important regulators for monocyte innate activation (Pekkari et al, **Blood**, 2005, 105:1598, Kong et al, **J Exp Med**, 2007, 204:2719), we therefore used these pathways to evaluate the activation status of HCC-SN-exposed monocytes. Furthermore, in one of our previous studies (Kuang et al, **J Immunol**, 2008, 181:3089), we have observed that the semimature DCs generated in tumor microenvironments primarily activated autologous T cells and subsequently led to apoptosis of those cells. It is well known that CD95-CD95L interaction plays a crucial role in DC-mediated activation-induced T cell death (Strasser et al, **Immunity**, 2009, 30:180, Krammer et al, **Nature**, 2000, 12:789). Thus, we analyzed whether CD95-CD95L interaction also participated in TDC-mediated activated Fc γ RII^{low/-} IL-10-producing B cell generation. We have added this information in the revised manuscript (**Page 8, Line 9-10; Page 9, Line 21-25**).

Reply to Reviewer 3

The most important concern is the lack of mentioning/describing/testing of the different isoforms of FcγRII. In humans, at least two isoforms of FcγRII are present. FcγRIIa has been described as a potent immune-activating receptor and it contains an ITAM motif, capable of mediating phagocytosis, ADCC, and initiation of inflammatory cytokine release. FcγRIIb has an ITIM motif and is an immune-inhibiting receptor. Both isoforms have been shown to be present on human B cells. Unfortunately, the monoclonal antibody used 6C4, specifically recognizes the two isoforms of human FcγRII. Therefore, the contribution of either one of the isoforms can not be distinguished nor commented on.

Reply to comment: In the current study, we only used down-regulated FcγRII as a marker for B cell activation. Correspondingly, the FcγRII^{low/-} B cells from HCC tissues exhibited a BTLA⁻CD69⁺ phenotype, confirming the activated form of cells. We afterward demonstrated that TDCs promoted B cell activation and subsequent IL-10 production via CD95-CD95L interaction. Therefore, down-regulated FcγRII is only a marker for B cell activation in our system, and FcγRII signals did not participate in B cell activation and subsequent IL-10 production. Notably, using real-time PCR, we found that, during TDC-mediated B cell activation, all FcγRII isoforms FcγRIIa and FcγRIIb were down-regulated (**Fig. 4 for Reviewer**).

Fig. 4 for Reviewer: Effect of TDCs on autologous blood B cell FcγRIIa and FcγRIIb expression.

Blood B cells were left untreated or cultured with TDCs for 2 days. Thereafter, the B cells were sorted and the expression of FcγRIIa and FcγRIIb in these cells was determined by real-time PCR. Results represent three independent experiments (n = 3).

Major comments:

1. The authors recently published another novel regulatory B cell population that promotes disease progression in hepatoma. This population expressed high levels of PD-1. Is this the same population? Are the FcγRII^{low/-} B cells also expressing PD-1?

Reply: The issue raised by the Reviewer is very attractive. We accordingly analyzed the expression of FcγRII and PD-1 in HCC-infiltrating B cells during this round of revision. As shown in **Fig. 3 for Reviewer**, ~60% PD-1^{hi} B cells exhibited an FcγRII^{low/-} phenotype, whereas < 10% FcγRII^{low/-} B cells were PD-1^{hi}. Interestingly, the sorted PD-1^{hi}FcγRII^{low/-} B cells from HCC tissues, without PD-1 triggering, could not produce IL-10. By contrast, the PD-1^{low/-}FcγRII^{low/-} B cells, without additional stimulation, were the major source of IL-10 production in tumor B cells. Therefore, over 90% FcγRII^{low/-} B cells were not PD-1^{hi} B cells, and they acquired ability to produce IL-10 via a mechanism differed from PD-1^{hi} B cells.

2. The way DCs are defined in the manuscript is not precise. With immunohistochemistry DCs are simply defined as S100⁺, which may include a wide variety of cells. The expression of some S100 molecules has been associated with myeloid-derived suppressor cells (MDSCs), which are enriched in the tumor microenvironment. Further on, when these tumor DCs are sorted as CD45⁺CD15⁻CD11c^{hi}CD11b^{hi} cells, a definition that encompasses a lot of myeloid populations, including monocytes and MDSCs. The authors should be very careful how they call the cells they are using or simply use more markers to pinpoint the populations they are using. An example of the gating strategy should also be shown.

Reply: S100 is considered as marker for DCs (de Vries et al, **Nat Biotechnol**, 2005, 23:1407, Popov et al, **J Clin Invest**, 2006, 12:3160). We agree with the Reviewer that S100 is also weakly expressed by certain myeloid cell subpopulations in inflammatory diseases. As suggested by the Reviewer, we used immunohistochemical dual-staining of S100 and CD15 (marker for neutrophils), or S100 and CD68 (marker for monocytes/macrophages) to ascertain whether neutrophils or monocytes/macrophages in human HCC tissues expressed S100 molecule. The results showed that S100 antigen was negative or only weakly expressed on the tumor-infiltrating CD68⁺ or CD15⁺ cells (**Fig. 5 for Reviewer**) in HCC tissues. As shown in previous Supplementary Fig. 2, the sorted CD45⁺CD15⁻CD11c^{hi}CD11b^{hi} cells from HCC tissues were S100⁺, whereas the sorted CD45⁺CD15⁻CD11c⁻CD11b^{hi} cells were S100⁻. Thus, these data showed that, in human HCC tissues, S100 was not expressed by myeloid cells neutrophils and monocytes/macrophages.

Fig. 5 for Reviewer: Expression of S100, CD15, and CD68 in human HCC tissues. Frozen sections of hepatocellular carcinoma tissue were stained for S100 (red) and CD15 (green), or S100 (red) and CD68 (green). 1 of 8 representative samples is shown.

3. The authors are claiming that monocytes within the local tumor environment are transformed into semimature DCs. The authors have not mentioned anywhere in the manuscript how they are generating these monocyte-derived DCs? Is it with the classical protocol of ex vivo treatment with IL-4 and GM-CSF or are they spontaneously differentiating into DCs? If it is the first situation, making the claim that monocytes are differentiating into semimature DCs within the tumor environment is not at all correct, unless the authors prove that the tumor microenvironment contains IL-4 and GM-CSF. A very relevant question that comes here is what kind of effects does the tumor environment have on monocytes (without the additional treatment with IL-4 and GM-CSF) and most importantly on primary DCs (and not monocyte-derived DCs).

Reply: We apologize for the lack of methods about the generation of tumor-related DCs and have added this information into Methods section of the revised manuscript (**Page 18, Line 20-25; Page 19, Line 1-6**). To generate TMO-DCs, tumor-derived CD14⁺ monocytes were cultured for 6 days in complete RPMI medium supplemented with GM-CSF and IL-4. Half of the culture medium was replaced on day 3. Our previous data (Fig. 3a-c) have suggested that, if TDCs were derived from tumor-infiltrating monocytes, they would obtain a semimature phenotype and an ability to induce activated FcγRII^{low/-} IL-10-producing B cells. Consistent with this, the TDCs isolated directly from HCC tissues were semimature and effectively generated activated FcγRII^{low/-} IL-10-producing B cells (Fig. 2d-f).

At present, the source and regulating mechanism of tumor-infiltrating DCs are still unknown. As pointed out by the Reviewer, GM-CSF plus IL-4 is considered as a classical protocol for the generation monocyte-derived DCs. In fact, increased GM-CSF production by cancer cells have been found in several recent studies (Bayne et al, **Cancer Cell**, 2012, 21:822, Pylayeva-Gupta et al, **Cancer Cell**, 2012, 21:836, Su et al, **Cancer Cell**, 2014, 25:605). We also detected 10.5 ± 0.4 ng/ml GM-CSF in HCC-SN and only 0.9 ± 0.09 ng/ml

GM-CSF in LSN (unpublished data). Furthermore, Th2 cells are considered as major cellular source of IL-4. Although the absolute numbers of Th2 cells are not as pronounced as those of Th1, Th17, and Treg cells in tumor tissues, the existence of Th2 cells in human HCC tumors were observed (Budhu et al, **Cancer Cell**, 2006, 10:99). Therefore, the HCC environments have the potential to generate monocyte-derived DCs.

To further demonstrate the important role of TMO-derived DCs in activated $Fc\gamma RII^{low/-}$ IL-10-producing B cell generation, we have conducted the following experiments during this round of revision.

Fig. 6 for Reviewer: Effect of DCs on B cell activation and IL-10 production.

(a) Phenotypic features of HCC-SN-treated monocyte-derived DCs, normal DCs treated with HCC-SN or LPS. (b) Effect of different DCs on autologous B cell activation and IL-10 production. Results represent three independent experiments (n = 3).

(1) We first generated HCC-SN-treated monocyte-derived DCs (HCC-SN/DCs) and normal monocyte-derived DCs that were left untreated or treated with HCC-SN (DC+HCC-SN) or LPS (DC+LPS) for the final 1 day. HCC-SN-treated monocyte-derived DCs exhibited a semimature phenotype; normal monocyte-derived DCs exposed to HCC-SN acquired a mature phenotype as those displayed by LPS-induced mature DCs. Thus, early education of monocytes by HCC environments is critical for acquisition of a semimature phenotype of DCs (**Fig. 6 for Reviewer**). In other words, if tumor-infiltrating DCs migrate directly from normal tissues or lymphoid organ, they will acquire a full mature phenotype in HCC environments.

(2) We subsequently co-cultured these cells with autologous blood B cells as previously described. HCC-SN-treated monocyte-derived DCs, but not normal monocyte-derived DCs treated with HCC-SN or LPS, effectively promoted activated $Fc\gamma RII^{low/-}$ IL-10-producing B

cell generation (**Fig. 6 for Reviewer**). Therefore, early education of monocytes by HCC environment also determined the final function of DCs.

4. The way HCC supernatants are prepared doesn't preclude immune cell-derived cytokines and factors. If the pure effect of HCC cells is sought after, then these cells should be purified prior to supernatant preparation. As mentioned above, the effect of such supernatants on both pure monocytes and pure primary DCs should be determined.

Reply: As pointed out by the Reviewer, the primary HCC-SN was from the culture of completely digested HCC tumor biopsy specimens that contained immune cells. HCC-SN prepared using such a protocol can represent the complete HCC environments (Xiao et al, **Cancer Discov**, 2016, 6,546). We did not exclude the contribution of immune cell-derived cytokines and factors to tumor microenvironments. Moreover, we have previously found that culture supernatants derived from established hepatoma cell lines also effectively induced semimature DCs (Kuang et al, **J Immunol**, 2008, 181, 3089). Thus, we concluded that soluble factors derived from hepatoma cells are critical for semimature DC formation.

Furthermore, we have demonstrated that the monocytes/macrophages (TAMs) directly isolated from HCC tissues only weakly affected B cell activation and subsequent IL-10 production in previous version of the manuscript (Fig. 2e,f). As suggested by the Reviewer, we also determined the effect of HCC-SN on normal DC phenotype and function. The results showed that normal monocyte-derived DCs exposed to HCC-SN acquired a mature phenotype and did not affect the generation of activated FcγRII^{low/-} IL-10-producing B cells (**Fig. 6 for Reviewer**).

5. The semimature status of tumor-derived "DCs" might be the reason why B cells are activated and induced to produce IL-10. It is very important to compare those tumor-derived "DCs" with primary activated DCs (activated by example a TLR ligand). It is also relevant to determine the cytokine production profile of those tumor-derived "DCs" as such cytokines have major role in promoting B cell activation and antibody production and antibody class-switching.

Reply: As shown in **Fig. 6 for Reviewer**, TMO-derived DCs, but not normal monocyte-derived DCs treated with HCC-SN or LPS, effectively promoted the generation of activated FcγRII^{low/-} IL-10-producing B cells. In addition, we have previously shown that FcγRII^{low/-} B cells isolated from HCC tissues lost the ability to differentiate into plasma cells (Fig.1e) and FcγRII^{low/-} B cells generated in vitro by TDCs could not differentiate into plasma cells (Supplementary Fig. 2e).

Minor comments:

1. Figure 1b: how does FcγRII expression by B cells in healthy donors compare to that of HCC patients (both blood and healthy liver tissue). Moreover, does the same FcγRII expression pattern in HCC patients hold in other tissues (for example lymph nodes)?

Reply: As suggested by the Reviewer, we also determined the FcγRII expression on B cells from healthy blood (n = 5), normal liver (tissue distal to a liver hemangioma; n = 2), and non-tumor-draining lymph nodes from patients with gastric cancer (n = 3). As shown in **New Fig. 1b** and **Fig. 7 for Reviewer**, we only observed a marked down-regulation of FcγRII on B cell from HCC tissues.

Fig. 7 for Reviewer: Analysis of FcγRII expression on B cells derived from healthy blood (n = 5), HCC blood (n = 15), normal liver (n = 2), HCC tumor (n = 15), and non-tumor-draining lymph nodes from patients with gastric cancer (n = 3).

2. Do FcγRII⁺ B cells adopt the same profile as FcγRII^{low/-} B cells when they are activated?

Re: FcγRII^{low/-} B cells from HCC tissues mainly exhibited a CD69⁺BTLA⁻ phenotype in HCC tissues (Fig. 1c). In previous Fig. 2e, 3b,e, and 4a,b,i, we have shown that, when FcγRII⁺ B cells activated, they acquired a CD69⁺FcγRII^{low/-} phenotype. In the revised manuscript, we also found that FcγRII^{low/-} B cells generated in vitro are BTLA⁻ (**New Supplementary Fig. 2b**)

3. Figure 1e: what is the CD40 expression level by FcγRII^{low/-} B cells? The reason for no antibody production might be due to lack of proper stimulation.

Reply: We have previous found that CD40 was down-regulated on B cells from HCC tissues compared with paired blood samples (Fig. 2A in Xiao et al, **Cancer Discov**, 2016, 6,546). However, we detected similar CD40 expression on FcγRII^{low/-} B cells and FcγRII⁺ B cells from HCC tissues (**Fig. 8 for Reviewer**).

Fig. 8 for Reviewer: Analysis of CD40 expression on B cells derived from human HCC blood and tumor tissues.

Results represent three independent experiments (n = 3).

4. Figure 2c: the authors haven't defined what is considered as "high" S100⁺ DC density and what is "low" S100⁺ DC density.

Reply: To evaluate the effects of S100⁺ cells in different area of HCC tissues on patient's survival, the patients were divided into two groups according to the median value of S100⁺ DC densities in nontumoral, peritumoral stromal, and intratumoral regions of HCC. The median value of S100⁺ DC densities in nontumoral, peritumoral stromal, and intratumoral regions of HCC was listed in previous Supplementary Table 1. We have emphasized this information in the Figure Legend of the revised manuscript.

Reviewers' Comments:

Reviewer #1 (Remarks to the Author):

The authors have answered every my comment. The revised manuscripts reads well, and the results seem to be interpreted correctly. I have no more comments.

Reviewer #2 (Remarks to the Author):

Authors have addressed my concerns.

Minor comments:

Some grammatical mistakes are found in the manuscript.

E.g.

line 189: the conditioned medium from TDCs did promoted (promote) B cell activation and IL-10 production.

Please proof-read carefully.

The authors have included some figures in the response letter to support their findings. It will be nice to include those in the supplementary figures of the manuscript.

Reviewer #3 (Remarks to the Author):

The reply from the authors is satisfactory. No further comments.

Reply to Reviewer 2

Minor comments:

1. Some grammatical mistakes are found in the manuscript. E.g. line 189: the conditioned medium from TDCs did promoted (promote) B cell activation and IL-10 production. Please proof-read carefully.

Reply: We thank the Reviewer and have corrected the grammatical mistakes in the revised manuscript.

2. The authors have included some figures in the response letter to support their findings. It will be nice to include those in the supplementary figures of the manuscript.

Reply: We thank the Reviewer for the thoughtful suggestions and have included some previous Fig. 2, 4, 5, and 7 for Reviewer as **New Supplementary Fig. 1d, 2d, and 2a and Fig. 1b**, respectively, in the revised manuscript.

Considering that the journal will publish the reviewer comments to the authors and author rebuttal letters, we showed previous Fig. 1, 3, 6, and 8 for Reviewer only in the rebuttal letter to make our manuscript more readable.